# Global distribution of sporadic sapovirus infections: A systematic review and meta-analysis

**Marta Diez Valcarce[1,2]◉, Anita K. Kambhampati[1]◉, Laura E. Calderwood[1,3], Aron J. Hall[1], Sara A. Mirza[1], Jan Vinjé[ID][1] ***

**1** Division of Viral Diseases, Centers for Disease Control and Prevention, Atlanta, GA, United States of America, **2** Emory University Rollins School of Public Health, Atlanta, GA, United States of America, **3** Cherokee Nation Assurance, Arlington, VA, United States of America

◉ These authors contributed equally to this work.
* jvinje@cdc.gov

**Data Availability Statement:** All relevant data are within the manuscript and its S1 Checklist and S1 Appendix files.

## Abstract

Acute gastroenteritis (AGE), characterized by diarrhea and vomiting, is an important cause of global mortality, accounting for 9% of all deaths in children under five years of age. Since the reduction of rotavirus in countries that have included rotavirus vaccines in their national immunization programs, other viruses such as norovirus and sapovirus have emerged as more common causes of AGE. Due to widespread use of real-time RT-PCR testing, sapovirus has been increasingly reported as the etiologic agent in both AGE outbreaks and sporadic AGE cases. We aimed to assess the role of sapovirus as a cause of endemic AGE worldwide by conducting a systematic review of published studies that used molecular diagnostics to assess the prevalence of sapovirus among individuals with AGE symptoms. Of 106 articles included, the pooled sapovirus prevalence was 3.4%, with highest prevalence among children <5 years of age (4.4%) and among individuals in community settings (7.1%). Compared to studies that used conventional RT-PCR, RT-qPCR assays had a higher pooled prevalence (5.6%). Among individuals without AGE symptoms, the pooled sapovirus prevalence was 2.7%. These results highlight the relative contribution of sapovirus to cases of AGE, especially in community settings and among children <5 years of age.

## Introduction

Acute gastroenteritis (AGE), characterized by diarrhea and vomiting, remains an important cause of global morbidity and mortality accounting for 9% of all deaths in children under five years of age [1, 2]. The burden of disease is particularly high in low income countries in Africa and South Asia, where AGE causes more than a quarter of all deaths in children younger than five years of age and severe outcomes are frequent [2]. Since the widespread implementation of rotavirus vaccines, the relative impact of caliciviruses, which include norovirus and sapovirus, has increased [3–6]. In several countries where rotavirus vaccines have been

**Funding:** The author(s) received no specific funding for this work.

**Competing interests:** The authors have declared that no competing interests exist.

incorporated into national immunization programs, norovirus is now recognized as the most common cause of severe AGE in children <5 years of age [4], as well as a leading cause of all AGE globally [7]. Although historically less well documented, sapovirus is increasingly recognized as another common viral cause of sporadic AGE in young children as well as outbreaks involving contaminated food and person-to-person transmission among adults [8–11].

Sapovirus was first described in 1976 in human diarrheal specimens by direct observation of the virus using electron microscopy [12]. Since then, a wide range of hosts have been identified, including pigs, bats, dogs, sea lions, and mink [13]. Although early reports described sapovirus infections associated with less severe symptoms than norovirus and rotavirus [14], more recent studies have found that sapovirus infections can result in hospitalizations and severe dehydration [15, 16]. Poor cognitive outcomes in children have been associated with sapovirus infections in lower and middle income countries [17], and although the exact mechanism of this neurologic phenomenon is unclear, studies in mice suggest the possibility of a pathogen-derived intestinal dysbiosis affecting the gut-brain axis [18].

Historically, data on the prevalence of sapovirus have been scarce, but over the last decade, laboratory diagnostics for sapovirus have transitioned to more sensitive molecular methods and multi-pathogen platforms, resulting in increased detection rates [19, 20]. Widespread use of these platforms, including by large-scale multinational studies, have led to more widely available data on the burden and etiologies of diarrheal disease [21, 22]. In one such study conducted in 8 different countries, sapovirus was the most frequently detected virus among children with AGE, more often than rotavirus, adenovirus, norovirus, or astrovirus [22]. We conducted a systematic literature review and meta-analysis to assess the global prevalence of sapovirus.

## Methods

Methods used for this study were based on a previously conducted systematic review and meta-analysis for norovirus [7]. We conducted a literature search of the PubMed and Scopus databases to identify peer-reviewed studies published through July 2020 that included data on sapovirus prevalence among individuals with AGE, as defined by the study (hereafter referred to as "AGE cases"). As a secondary objective, we also sought to assess the burden of sapovirus among individuals without symptoms of AGE (hereafter referred to as "non-AGE controls") and included prevalence data from non-AGE controls where available in conjunction with data from AGE cases.

Literature search terms consisted of "sapovirus" or "sapoviruses" in any field; the search was limited to studies in humans published in English. Articles were selected using the Preferred Reporting Items for Systematic Reviews and Meta-Analyses (PRISMA) guidelines [23, 24] (Fig 1). Two independent reviewers screened titles and abstracts for relevance, and articles deemed relevant by either reviewer were assessed in full by both reviewers. For inclusion, articles had to contain primary data from studies that were conducted continuously for ≥1 year and used RT-PCR-based diagnostics to identify sapovirus in stool specimens from patients with AGE. Articles that did not include sapovirus prevalence estimates were excluded, including those that described sapovirus outbreaks, secondary data analysis, or environmental studies, and those that tested for calicivirus without differentiating between norovirus and sapovirus. We also excluded studies that only tested for sapovirus among immunocompromised patients (e.g., transplant recipients, HIV-positive patients), since prevalence among these patients may be overestimated. Finally, we excluded studies that tested for sapovirus only among pathogen-negative specimens (e.g., studies in which sapovirus testing was conducted only among rotavirus-negative specimens).

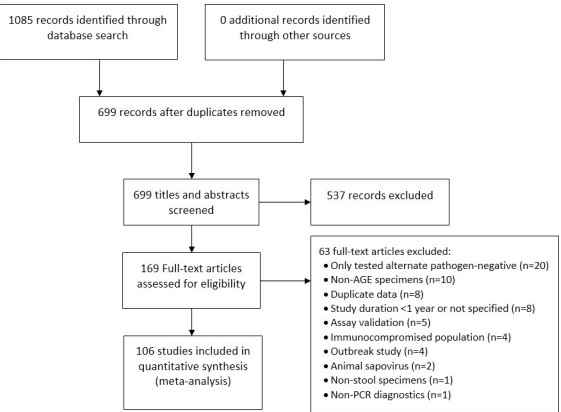

**Fig 1. PRISMA flow diagram for study selection process.**

In conjunction with a full text screen, independent double data abstraction was conducted using a standardized REDCap (Research Electronic Data Capture) database developed for the review [25]. Differences in data abstracted by the two reviewers were adjudicated by a third reviewer. The following data were abstracted from each article: first author, title, journal, year of publication, country, study duration, setting, method of detection (conventional RT-PCR or RT-qPCR, including RT-qPCR based multi-enteric pathogen platforms), number of AGE cases tested and positive for sapovirus (by age group and/or genogroup, if available), and number of non-AGE controls tested and positive for sapovirus (by age group and/or genogroup, if available). The number of sapovirus-positive cases and controls was calculated when only percentages were provided. To capture all sapovirus cases, those that were infected with both sapovirus and another AGE pathogen were included as sapovirus-positives.

Data were abstracted according to setting, which was used as a proxy for disease severity. Studies conducted in community settings ("community") were considered to represent patients with mild clinical illness; primary care facilities and outpatient clinics ("outpatient") were a proxy for patients with moderate illness; and emergency departments and inpatient hospital settings ("inpatient") represented severe illness. Studies for which a setting could not be determined (e.g., those that tested all stool specimens submitted to a national laboratory for AGE diagnostics) or for which data were not presented separately by setting were categorized as "mixed". Age-stratified data were abstracted as presented and collapsed into <5 year, ≥5 year, and "mixed" (all/unspecified ages) age categories. Data reported for children "≤5 years" or "≤60 months" were categorized into the <5 years age group. Countries in which studies were performed were also classified into three "development index" categories. Based on the United Nations Development Programme Human Development Index (HDI) [26], countries were first classified as "developed" (HDI >0.80) or "developing" (HDI <0.80). Developing countries were further divided into "low-mortality developing" (LMD) "high-mortality developing" (HMD) based on child mortality rates generated by the United Nations Interagency Group for Child Mortality Estimation (https://data.worldbank.org/indicator/SH.DYN.MORT).

Raw proportions were transformed using the Freeman-Tukey double arcsine transformation and pooled prevalence estimates were generated by fitting random-effects meta-analysis models using the inverse variance method for pooling, restricted maximum-likelihood estimation, and the Knapp and Hartung adjustment. The presence of bias in effect-size estimates was determined using Egger's regression test. Heterogeneity was assessed using the $I^2$ statistic. Subgroup analyses were conducted to assess differences in prevalence by age, setting, development

index, and method of detection. Analyses were conducted using the "meta" package in R (version 4.0.2, R Core Team, 2020) [27]; Fig 2 was created using the "ggplot" package. In addition to the primary meta-analyses, sapovirus genogroup data were summarized qualitatively due to limited available data.

## Results

The literature search yielded 1,085 articles, of which 169 full-text articles were assessed for inclusion (Fig 1). In total, data from 106 articles were included (Table 1 and S1 Appendix). The articles included data from 35 countries (Fig 2), among which 19 (54%) countries were represented by one study each. Among the 106 articles, the highest number of studies were conducted in China (17; 16%) followed by Japan (13, 12%) and the United States (12, 11%). High mortality developing countries were represented in 13 (12%) studies. All studies were conducted between 1990–2019 and had a median study duration of 2 years (range 1–10 years). Studies most frequently included data from children <5 years of age (64, 60%) and from inpatient settings (48, 45%).

Overall, the estimated prevalence of sapovirus among AGE cases was 3.4% (95% confidence interval [CI]: 2.8–4.1%, $I^2$: 97%) (Table 2). No evidence of bias in effect-size estimates was found in the studies overall (p = 0.09). There were significant differences in prevalence by age (p<0.01, residual $I^2$: 96%). The highest prevalence was found among children <5 years of age (4.4%, 95% CI 3.4–5.5%, $I^2$: 95%) compared to individuals ≥5 years of age (1.9%, 95% CI: 1.0%–3.1%, $I^2$: 86%). There were also significant differences by setting (p<0.01, residual $I^2$: 96%) with sapovirus prevalence highest in community settings (7.1%, 95% CI: 4.2–10.6%, $I^2$: 95%) followed by outpatient settings (4.0%, 95% CI: 2.6–5.7%, $I^2$: 98%), and lowest in studies conducted in inpatient settings (2.3%, 95% CI: 1.5–3.2%, $I^2$: 96%).

Sapovirus prevalence among AGE cases was also significantly different by method of detection (p<0.01, residual $I^2$: 97%) (Table 2). Studies utilizing RT-qPCR assays as the method of detection had a pooled prevalence of 5.6% (95% CI: 4.2–7.1%, $I^2$: 97%). In comparison, studies using conventional RT-PCR assays had a pooled prevalence of 2.5% (95% CI: 1.9–3.1%, $I^2$: 96%). No significant differences or reduction in heterogeneity was seen in prevalence among AGE cases when examined by development index (p = 0.23, residual $I^2$: 97%).

Among the 106 articles, 19 (17.9%) included data on sapovirus in non-AGE controls with a pooled prevalence of 2.7% (95% CI: 1.6–4.0%, $I^2$: 86%) (Table 3). In these studies, there were significant differences in prevalence by age (p<0.01, residual $I^2$: 79%) with a prevalence of 3.5% (95% CI: 2.2–5.0%, residual $I^2$: 83%) among children <5 years of age and 0.8% (95% CI: 0.0–2.2%, residual $I^2$: 35%) among individuals aged ≥5 years. Sapovirus prevalence among

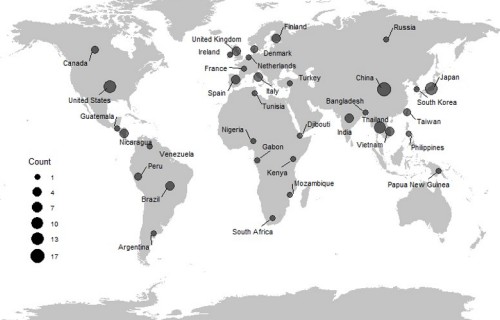

**Fig 2. Map of countries in which studies included in sapovirus meta-analysis were conducted.**

**Table 1. Characteristics of studies included in sapovirus meta-analysis.**

| | Number of studies[a] | Median number tested (range) | Median number positive (range) |
|---|---|---|---|
| **AGE cases** | 106 | 529 (72–13231) | 14 (0–559) |
| **Age[b]** | | | |
| <5 years | 64 | 471 (67–11800) | 16 (0–559) |
| ≥5 years | 21 | 392 (14–776) | 7 (0–42) |
| Mixed/unknown age | 38 | 553 (8–13231) | 11 (0–417) |
| **Setting[b]** | | | |
| Community | 13 | 357 (81–2422) | 23 (2–132) |
| Outpatient | 31 | 728 (123–3832) | 23 (2–417) |
| Inpatient | 48 | 459 (45–9597) | 8 (0–238) |
| Mixed/unknown setting | 24 | 802 (103–13231) | 39 (0–559) |
| **Method of Detection** | | | |
| Conventional RT-PCR | 70 | 580 (75–9597) | 11 (0–238) |
| RT-qPCR | 36 | 486 (102–13231) | 30 (1–559) |
| **Development Index** | | | |
| Developed | 48 | 697 (72–13231) | 21 (0–559) |
| Low mortality developing (LMD) | 45 | 491 (81–3832) | 9 (0–118) |
| High mortality developing (HMD) | 13 | 350 (75–3099) | 18 (0–238) |

[a]Of the 106 total studies with data on AGE cases, 19 studies also included data on non-AGE controls. The median number of controls tested was 300 (range: 19-2205) and a median of 8 controls (range: 0-73) were sapovirus positive.

[b]Multiple age categories or settings were represented in certain studies.

non-AGE controls also differed significantly by setting (p<0.01, residual $I^2$: 86%) with the highest prevalence in the community setting (3.5% 95% CI: 1.8–5.7, $I^2$: 91%). There were no significant differences in the prevalence among non-AGE controls or reduction in heterogeneity when stratified by method of detection (p = 0.17, residual $I^2$: 82%) or by development index (p = 0.50, residual $I^2$: 86%) (Table 3).

Sapovirus genogroup data for AGE cases were available in 47 (44%) studies. GI sapoviruses were the most commonly reported genogroup and were reported in all but one study with available genogroup data, with a detection rate between 15–100% among typed specimens. GII sapoviruses were reported in 39 studies, with detection ranging between 2–65% of typed specimens. GIV and GV sapoviruses were reported in 16 studies and 9 studies, respectively, with detection rates ranging from 3–100% for GIV and 1–14% for GV. GI and GII sapoviruses were detected in all 6 studies with available genogroup data for non-AGE controls, with similar detection rates of 28–71% for GI and 29–67% for GII. Two studies reported GIV sapoviruses among non-AGE controls with estimates of 6% and 33% among typed specimens from non-AGE controls. One study reported GV sapovirus in 19% of typed specimens from non-AGE controls.

## Discussion

Sapovirus has increasingly been recognized as a cause of AGE in young children. We performed a systematic review and meta-analysis of more than 100 studies which found that sapovirus was detected in approximately 3.4% of all AGE cases regardless of detection method and 5.6% of AGE cases when RT-qPCR assays were used. Further, although increasingly identified as an etiology of AGE, we found that sapovirus was detected in 2.7% of non-AGE controls.

These estimates represent sapovirus data from 35 countries, with no significant differences observed by development index. However, over half of countries were represented by

**Table 2. Prevalence of sapovirus among AGE cases, overall and stratified by setting, method of detection, and development index.**

|  | Prevalence of sapovirus among AGE cases [95% confidence interval] | p-value |
|---|---|---|
| **Overall** | 3.4 [2.8–4.1] | – |
| **Age** |  | <0.01 |
| <5 years | 4.4 [3.4–5.5] |  |
| ≥5 years | 1.9 [1.0–3.1] |  |
| Mixed | 2.7 [1.9–3.7] |  |
| **Setting** |  | <0.01 |
| Community | 7.1 [4.2–10.6] |  |
| Outpatient | 4.0 [2.6–5.7] |  |
| Inpatient | 2.3 [1.5–3.2] |  |
| Mixed | 4.0 [3.0–5.0] |  |
| **Method of detection** |  | <0.01 |
| Conventional RT-PCR | 2.5 [1.9–3.1] |  |
| RT-qPCR | 5.6 [4.2–7.1] |  |
| **Development Index** |  | 0.23 |
| Developed | 4.0 [3.0–5.0] |  |
| LMD | 2.9 [1.9–4.0] |  |
| HMD | 4.2 [2.8–6.0] |  |

only one study, and few studies included data from developing countries. Since diarrheal diseases can cause substantial morbidity and mortality, to provide more accurate global estimates, additional studies are needed in underrepresented countries to more fully characterize the proportion of AGE associated with sapovirus. Improved surveillance for sapovirus in developing countries may provide a more comprehensive understanding of global sapovirus distribution.

**Table 3. Prevalence of sapovirus among non-AGE controls, overall and stratified by setting, method of detection, and development index.**

|  | Prevalence of sapovirus among non-AGE controls [95% confidence interval] | p-value |
|---|---|---|
| **Overall** | 2.7 [1.6–4.0] | – |
| **Age** |  | <0.01 |
| <5 years | 3.5 [2.2–5.0] |  |
| ≥5 years | 0.8 [0.0–2.2] |  |
| Mixed | 0.8 [0.5–1.1] |  |
| **Setting** |  | <0.01 |
| Community | 3.5 [1.8–5.7] |  |
| Outpatient | 0.0 [0.0–5.9] |  |
| Inpatient | 2.5 [0.7–5.3] |  |
| Mixed | 3.1 [0.6–7.2] |  |
| **Method of detection** |  | 0.17 |
| Conventional RT-PCR | 2.1 [0.4–5.0] |  |
| RT-qPCR | 3.8 [2.5–5.3] |  |
| **Development Index** |  | 0.50 |
| Developed | 2.4 [1.7–3.3] |  |
| LMD | 3.8 [1.4–7.3] |  |
| HMD | 2.8 [0.0–20.2] |  |

Sapovirus prevalence varied significantly by setting, with the highest pooled prevalence in community settings, followed by outpatient settings, and the lowest prevalence in inpatient settings. A similar gradient of decreasing prevalence has been observed for norovirus [8], and supports the understanding that sapovirus generally causes mild, self-limiting illness. However, more severe clinical symptoms such as severe dehydration requiring medical attention and/or hospitalization have been reported [15, 28].

For norovirus and rotavirus [29, 30], it has been well documented that children <5 years of age have a higher prevalence compared to older individuals. Similarly, the prevalence of sapovirus among AGE cases, as well as among non-AGE controls, was highest in children <5 years of age. These results suggest that infections during childhood may provide some protection against subsequent infections in adulthood, a promising sign for potential vaccines. However, the degree to which reinfection with a specific genotype provides cross-protection against other genotypes or genogroups requires additional studies [31].

While we did not find significant differences in observed sapovirus prevalence by method of detection among non-AGE controls, studies that utilized RT-qPCR assays yielded a significantly higher prevalence among AGE cases than those that utilized conventional RT-PCR assays. These findings are not surprising since the addition of fluorescent probes in RT-qPCR assays increases the diagnostic specificity of these assays [32]. The studies in this meta-analysis were conducted during the last 30 years with more recent studies using the more sensitive RT-qPCR methods [33] including RT-qPCR-based multi-enteric pathogen platforms [34]. Due to the rapid turnaround time and the ability to detect several pathogens in a single test, these platforms are becoming increasingly popular in large-scale AGE surveillance studies [19, 34] and therefore, an updated meta-analysis for sapovirus may be necessary in the future.

Our study has several limitations. Although age (<5 years), method of detection, and community setting were significantly associated with increased prevalence of sapovirus, accounting for these factors did not result in a substantial decrease in heterogeneity, suggesting there may be other factors that affect the wide range of sapovirus prevalence observed globally. While we restricted our analyses to studies that tested for sapovirus among a defined population of patients with AGE symptoms and excluded those that screened out specimens that tested positive for other AGE pathogens, there may have been differences in the AGE case definition used by each study, thus potentially contributing to the observed heterogeneity. Further, although we attempted to account for differences in access to healthcare and water and sanitation conditions by examining prevalence by country development index, this grouping does not fully capture the spectrum of access on an individual level. Finally, the studies included in this analysis were conducted over a nearly 30-year time span. Changing patterns of virus circulation due to factors such as the introduction of rotavirus vaccines [15], climate patterns [35], and socio-demographic changes may also have contributed to differences in sapovirus prevalence.

The results of this study highlight the prevalence of sporadic sapovirus infections among AGE cases, in particular among children <5 years of age. Additional data on the burden of sapovirus, particularly among individuals without AGE symptoms, as well as data on circulating sapovirus genotypes, especially in settings of increased AGE-associated morbidity and mortality, are important to understand etiologies of AGE and inform strategies for prevention and control.

## Supporting information

**S1 Appendix. List of studies included in sapovirus meta-analysis.**
(DOCX)

**S1 Checklist. PRISMA checklist.**
(DOCX)

## Acknowledgments

**Disclaimer:** The findings and conclusions in this article are those of the authors and do not necessarily represent the official position of the Centers for Disease Control and Prevention.

## Author Contributions

**Conceptualization:** Jan Vinjé.

**Data curation:** Marta Diez Valcarce, Anita K. Kambhampati.

**Formal analysis:** Marta Diez Valcarce, Anita K. Kambhampati.

**Methodology:** Marta Diez Valcarce, Anita K. Kambhampati, Laura E. Calderwood.

**Project administration:** Jan Vinjé.

**Software:** Anita K. Kambhampati.

**Supervision:** Aron J. Hall, Sara A. Mirza, Jan Vinjé.

**Validation:** Marta Diez Valcarce, Anita K. Kambhampati, Laura E. Calderwood.

**Visualization:** Marta Diez Valcarce, Anita K. Kambhampati.

**Writing – original draft:** Marta Diez Valcarce.

**Writing – review & editing:** Anita K. Kambhampati, Laura E. Calderwood, Aron J. Hall, Sara A. Mirza, Jan Vinjé.

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
