## [Editor Report · Decision Letter 0]

3 Jun 2021

PONE-D-21-11650

Global distribution of sporadic sapovirus infections: a systematic review and meta-analysis

PLOS ONE

Dear Dr. Vinjé,

Thank you for submitting your manuscript to PLOS ONE. After careful consideration, we feel that it has merit but does not fully meet PLOS ONE’s publication criteria as it currently stands. Therefore, we invite you to submit a revised version of the manuscript that addresses the points raised during the review process.

The manuscript focuses on a very important issue, and is, overall, methodologically appropriate. I have a few reservations about the manuscript as it currently stands. However, if the authors will be able to address the following issues, I believe that it may be re-considered for publication.

1. In addition to reporting the results (and commenting in them) throughout the text, the pooled results (at leeast those referring to the main analyses) should be displayed in appropriate tables. This would substantially increase the clarity of the manuscript, and help the readers delving into data.

2. The use of "cases" and "controls" definition throughout the text is potentially misleading to define subjects with and without AGE, as there is no formal comparison between groups. I would replace with something like "symptomatic" and "asymptomatic".

3. Meta-regression - partially linked to point 1: which studies were included? And which are the covariates considered? Please add some more details and display the results of the meta-regression in a specific table. 

We look forward to receiving your revised manuscript.

Kind regards,

Maria Elena Flacco, M.D.

Academic Editor

PLOS ONE

Journal Requirements:

2. We note that Figure 2 in your submission contain map images which may be copyrighted. All PLOS content is published under the Creative Commons Attribution License (CC BY 4.0), which means that the manuscript, images, and Supporting Information files will be freely available online, and any third party is permitted to access, download, copy, distribute, and use these materials in any way, even commercially, with proper attribution. For these reasons, we cannot publish previously copyrighted maps or satellite images created using proprietary data, such as Google software (Google Maps, Street View, and Earth). For more information, see our copyright guidelines: http://journals.plos.org/plosone/s/licenses-and-copyright.

2.1.    You may seek permission from the original copyright holder of Figure 2 to publish the content specifically under the CC BY 4.0 license. 

2.2.    If you are unable to obtain permission from the original copyright holder to publish these figures under the CC BY 4.0 license or if the copyright holder’s requirements are incompatible with the CC BY 4.0 license, please either i) remove the figure or ii) supply a replacement figure that complies with the CC BY 4.0 license. Please check copyright information on all replacement figures and update the figure caption with source information. If applicable, please specify in the figure caption text when a figure is similar but not identical to the original image and is therefore for illustrative purposes only.

3. We note that this manuscript is a systematic review or meta-analysis; our author guidelines therefore require that you use PRISMA guidance to help improve reporting quality of this type of study. Please upload copies of the completed PRISMA checklist as Supporting Information with a file name “PRISMA checklist”.

---

## [Author Response · Author response to Decision Letter 0]

13 Jul 2021

PONE-D-21-11650

RESPONSE TO REVIEWERS

The manuscript focuses on a very important issue, and is, overall, methodologically appropriate. I have a few reservations about the manuscript as it currently stands. However, if the authors will be able to address the following issues, I believe that it may be re-considered for publication.

Thank you very much for your review of the paper. We have made the revisions below. In addition to the suggested revisions, we have separated the “method of detection” category into: 1. detection methods based on conventional PCR and 2. Detection methods based on real time PCR (which includes probe-based realtime RT-PCR methods as well as modern multienteric pathogen platforms such as FilmArray (Biofire)). We believe that these two categories account more accurately the potential differences observed among the published studies included in our analysis. 

1. In addition to reporting the results (and commenting in them) throughout the text, the pooled results (at least those referring to the main analyses) should be displayed in appropriate tables. This would substantially increase the clarity of the manuscript, and help the readers delving into data.

As requested, we have included the pooled results in a table, which also includes data that were previously presented in supplementary figures. We have removed the original figures.

2. The use of "cases" and "controls" definition throughout the text is potentially misleading to define subjects with and without AGE, as there is no formal comparison between groups. I would replace with something like "symptomatic" and "asymptomatic".

Thank you for this feedback. We have revised our terminology to indicate these groups refer to “AGE cases” and “non-AGE controls” and clarified in the methods that the terminology refers to the case definitions used in the original studies, many of which had formal comparisons. Studies had varying definitions for both AGE cases and non-AGE controls; for example, in some studies, non-AGE controls may have been completely asymptomatic for a specified period of time, while others allowed for inclusion as controls those who had some symptoms but did not meet the AGE case definition. Thus, it may not be appropriate to refer to these groups as symptomatic/asymptomatic. 

3. Meta-regression - partially linked to point 1: which studies were included? And which are the covariates considered? Please add some more details and display the results of the meta-regression in a specific table. 

The meta-regression analysis was conducted as a sensitivity analysis and did not add additional clarity to the other results. As such, we have removed the description of the meta-regression from our manuscript.

 

Journal Requirements:

Please review your reference list to ensure that it is complete and correct. 

We have confirmed that the manuscript meets PLOS ONE’s style requirements and updated any elements as required. 

2. We note that Figure 2 in your submission contain map images which may be copyrighted. All PLOS content is published under the Creative Commons Attribution License (CC BY 4.0), which means that the manuscript, images, and Supporting Information files will be freely available online, and any third party is permitted to access, download, copy, distribute, and use these materials in any way, even commercially, with proper attribution. For these reasons, we cannot publish previously copyrighted maps or satellite images created using proprietary data, such as Google software (Google Maps, Street View, and Earth). For more information, see our copyright guidelines: http://journals.plos.org/plosone/s/licenses-and-copyright. We require you to either (1) present written permission from the copyright holder to publish these figures specifically under the CC BY 4.0 license, or (2) remove the figures from your submission.

Thank you. This figure was created in R using the ggplot package, which has been cited in the text (lines 121-122). 

3. We note that this manuscript is a systematic review or meta-analysis; our author guidelines therefore require that you use PRISMA guidance to help improve reporting quality of this type of study. Please upload copies of the completed PRISMA checklist as Supporting Information with a file name “PRISMA checklist”.

The PRISMA checklist has been re-uploaded. The file name has been updated to “PRISMA Checklist”.

---

## [Editor Report · Decision Letter 1]

16 Jul 2021

Global distribution of sporadic sapovirus infections: a systematic review and meta-analysis

PONE-D-21-11650R1

Dear Dr. Vinjé,

We’re pleased to inform you that your manuscript has been judged scientifically suitable for publication and will be formally accepted for publication once it meets all outstanding technical requirements.

Kind regards,

Maria Elena Flacco, M.D.

Academic Editor

PLOS ONE
---

## [Editor Report · Acceptance letter]

4 Aug 2021

PONE-D-21-11650R1 

Global distribution of sporadic sapovirus infections: a systematic review and meta-analysis 

Dear Dr. Vinjé:

I'm pleased to inform you that your manuscript has been deemed suitable for publication in PLOS ONE. Congratulations! Your manuscript is now with our production department. 

Kind regards, 

on behalf of

Dr. Maria Elena Flacco 

Academic Editor

PLOS ONE